# Approximate Supermodularity Bounds for Experimental Design

**Luiz F. O. Chamon and Alejandro Ribeiro**
Electrical and Systems Engineering
University of Pennsylvania
{luizf,aribeiro}@seas.upenn.edu

## Abstract

This work provides performance guarantees for the greedy solution of experimental design problems. In particular, it focuses on A- and E-optimal designs, for which typical guarantees do not apply since the mean-square error and the maximum eigenvalue of the estimation error covariance matrix are not supermodular. To do so, it leverages the concept of approximate supermodularity to derive non-asymptotic worst-case suboptimality bounds for these greedy solutions. These bounds reveal that as the SNR of the experiments decreases, these cost functions behave increasingly as supermodular functions. As such, greedy A- and E-optimal designs approach $(1 - e^{-1})$-optimality. These results reconcile the empirical success of greedy experimental design with the non-supermodularity of the A- and E-optimality criteria.

## 1 Introduction

Experimental design consists of selecting which experiments to run or measurements to observe in order to estimate some variable of interest. Finding good designs is an ubiquitous problem with applications in regression, semi-supervised learning, multivariate analysis, and sensor placement [1–10]. Nevertheless, selecting a set of $k$ experiments that optimizes a generic figure of merit is NP-hard [11, 12]. In some situations, however, an approximate solution with optimality guarantees can be obtained in polynomial time. For example, this is possible when the cost function possesses a diminishing returns property known as *supermodularity*, in which case greedy search is near-optimal. Greedy solutions are particularly attractive for large-scale problems due to their iterative nature and because they have lower computational complexity than typical convex relaxations [11, 12].

Supermodularity, however, is a stringent condition not met by important performance metrics. For instance, it is well-known that neither the mean-square error (MSE) nor the maximum eigenvalue of the estimation error covariance matrix are supermodular [1, 13, 14]. Nevertheless, greedy algorithms have been successfully used to minimize these functions despite the lack of theoretical guarantees. The goal of this paper is to reconcile these observations by showing that these figures of merit, used in A- and E-optimal experimental designs, are approximately supermodular. To do so, it introduces different measures of approximate supermodularity and derives near-optimality results for these classes of functions. It then bounds how much the MSE and the maximum eigenvalue of the error covariance matrix violate supermodularity, leading to performance guarantees for greedy A- and E-optimal designs. More to the point, the main results of this work are:

1. The greedy solution of the A-optimal design problem is within a multiplicative $(1 - e^{-\alpha})$ factor of the optimal with $\alpha \geq [1 + \mathcal{O}(\gamma)]^{-1}$, where $\gamma$ upper bounds the signal-to-noise ratio (SNR) of the experiments (Theorem 3).

2. The value of the greedy solution of an E-optimal design problem is at most $(1 - e^{-1})(f(\mathcal{D}^\star) + k\epsilon)$, where $\epsilon \leq \mathcal{O}(\gamma)$ (Theorem 4).

3. As the SNR of the experiments decreases, the performance guarantees for greedy A- and E-optimal designs approach the classical $1 - 1/e$.

This last observation is particularly interesting since careful selection of experiments is more important in low SNR scenarios. In fact, unless experiments are highly correlated, designs have similar performances in high SNR. Also, note that the guarantees in this paper are not asymptotic and hold in the worst-case, i.e., hold for problems of any dimension and for designs of any size.

**Notation** Lowercase boldface letters represent vectors ($\boldsymbol{x}$), uppercase boldface letters are matrices ($\boldsymbol{X}$), and calligraphic letters denote sets/multisets ($\mathcal{A}$). We write $\#\mathcal{A}$ for the cardinality of $\mathcal{A}$ and $\mathcal{P}(\mathcal{B})$ to denote the collection of all finite multisets of the set $\mathcal{B}$. To say $\boldsymbol{X}$ is a positive semi-definite (PSD) matrix we write $\boldsymbol{X} \succeq 0$, so that for $\boldsymbol{X}, \boldsymbol{Y} \in \mathbb{R}^{n \times n}$, $\boldsymbol{X} \preceq \boldsymbol{Y} \Leftrightarrow \boldsymbol{b}^T \boldsymbol{X} \boldsymbol{b} \leq \boldsymbol{b}^T \boldsymbol{Y} \boldsymbol{b}$, for all $\boldsymbol{b} \in \mathbb{R}^n$. Similarly, we write $\boldsymbol{X} \succ 0$ when $\boldsymbol{X}$ is positive definite.

## 2 Optimal experimental design

Let $\mathcal{E}$ be a pool of possible experiments. The outcome of experiment $e \in \mathcal{E}$ is a multivariate measurement $\boldsymbol{y}_e \in \mathbb{R}^{n_e}$ defined as

$$\boldsymbol{y}_e = \boldsymbol{A}_e \boldsymbol{\theta} + \boldsymbol{v}_e, \tag{1}$$

where $\boldsymbol{\theta} \in \mathbb{R}^p$ is a parameter vector with a prior distribution such that $\mathbb{E}[\boldsymbol{\theta}] = \bar{\boldsymbol{\theta}}$ and $\mathbb{E}(\boldsymbol{\theta} - \bar{\boldsymbol{\theta}})(\boldsymbol{\theta} - \bar{\boldsymbol{\theta}})^T = \boldsymbol{R}_\theta \succ 0$; $\boldsymbol{A}_e$ is an $n_e \times p$ observation matrix; and $\boldsymbol{v}_e \in \mathbb{R}^{n_e}$ is a zero-mean random variable with arbitrary covariance matrix $\boldsymbol{R}_e = \mathbb{E}\, \boldsymbol{v}_e \boldsymbol{v}_e^T \succ 0$ that represents the experiment uncertainty. The $\{\boldsymbol{v}_e\}$ are assumed to be uncorrelated across experiments, i.e., $\mathbb{E}\, \boldsymbol{v}_e \boldsymbol{v}_f^T = \boldsymbol{0}$ for all $e \neq f$, and independent of $\boldsymbol{\theta}$. These experiments aim to estimate

$$\boldsymbol{z} = \boldsymbol{H}\boldsymbol{\theta}, \tag{2}$$

where $\boldsymbol{H}$ is an $m \times p$ matrix. Appropriately choosing $\boldsymbol{H}$ is important given that the best experiments to estimate $\boldsymbol{\theta}$ are not necessarily the best experiments to estimate $\boldsymbol{z}$. For instance, if $\boldsymbol{\theta}$ is to be used for classification, then $\boldsymbol{H}$ can be chosen so as to optimize the design with respect to the output of the classifier. Alternatively, transductive experimental design can be performed by taking $\boldsymbol{H}$ to be a collection of data points from a test set [6]. Finally, $\boldsymbol{H} = \boldsymbol{I}$, the identity matrix, recovers the classical $\boldsymbol{\theta}$-estimation case.

The experiments to be used in the estimation of $\boldsymbol{z}$ are collected in a multiset $\mathcal{D}$ called a *design*. Note that $\mathcal{D}$ contains elements of $\mathcal{E}$ with repetitions. Given a design $\mathcal{D}$, it is ready to compute an optimal Bayesian estimate $\hat{\boldsymbol{z}}_\mathcal{D}$. The estimation error of $\hat{\boldsymbol{z}}_\mathcal{D}$ is measured by the error covariance matrix $\boldsymbol{K}(\mathcal{D})$. An expression for the estimator and its error matrix in terms of the problem constants is given in the following proposition.

**Proposition 1** (Bayesian estimator)**.** *Let the experiments be defined as in* (1)*. For* $\boldsymbol{M}_e = \boldsymbol{A}_e^T \boldsymbol{R}_e^{-1} \boldsymbol{A}_e$ *and a design* $\mathcal{D} \in \mathcal{P}(\mathcal{E})$*, the unbiased affine estimator of* $\boldsymbol{z}$ *with the smallest error covariance matrix in the PSD cone is given by*

$$\hat{\boldsymbol{z}}_\mathcal{D} = \boldsymbol{H} \left[ \boldsymbol{R}_\theta^{-1} + \sum_{e \in \mathcal{D}} \boldsymbol{M}_e \right]^{-1} \left[ \sum_{e \in \mathcal{D}} \boldsymbol{A}_e^T \boldsymbol{R}_e^{-1} \boldsymbol{y}_e + \boldsymbol{R}_\theta^{-1} \bar{\boldsymbol{\theta}} \right]. \tag{3}$$

*The corresponding error covariance matrix* $\boldsymbol{K}(\mathcal{D}) = \mathbb{E}\left[ (\boldsymbol{z} - \hat{\boldsymbol{z}}_\mathcal{D})(\boldsymbol{z} - \hat{\boldsymbol{z}}_\mathcal{D})^T \mid \boldsymbol{\theta}, \{\boldsymbol{M}_e\}_{e \in \mathcal{D}} \right]$ *is given by the expression*

$$\boldsymbol{K}(\mathcal{D}) = \boldsymbol{H} \left[ \boldsymbol{R}_\theta^{-1} + \sum_{e \in \mathcal{D}} \boldsymbol{M}_e \right]^{-1} \boldsymbol{H}^T. \tag{4}$$

*Proof.* See extended version [15]. $\qquad\square$

The experimental design problem consists of selecting a design $\mathcal{D}$ of cardinality at most $k$ that minimizes the overall estimation error. This can be explicitly stated as the problem of choosing $\mathcal{D}$

with $\#\mathcal{D} \le k$ that minimizes the error covariance $\boldsymbol{K}(\mathcal{D})$ whose expression is given in (4). Note that (4) can account for unregularized (non-Bayesian) experimental design by removing $\boldsymbol{R}_\theta$ and using a pseudo-inverse [16]. However, the error covariance matrix is no longer monotone in this case—see Lemma 1. Providing guarantees for this scenario is the subject of future work.

The minimization of the PSD matrix $\boldsymbol{K}(\mathcal{D})$ in experimental design is typically attempted using scalarization procedures generically known as alphabetical design criteria, the most common of which are A-, D-, and E-optimal design [17]. These are tantamount to selecting different figures of merit to compare the matrices $\boldsymbol{K}(\mathcal{D})$. Our focus in this paper is mostly on A- and E-optimal designs, but we also consider D-optimal designs for comparison. A design $\mathcal{D}$ with $k$ experiments is said to be A-optimal if it minimizes the estimation MSE which is given by the trace of the covariance matrix,

$$\underset{|\mathcal{D}| \le k}{\text{minimize}} \quad \text{Tr}\left[\boldsymbol{K}(\mathcal{D})\right] - \text{Tr}\left[\boldsymbol{H}\boldsymbol{R}_\theta\boldsymbol{H}^T\right] \tag{P-A}$$

Notice that is customary to say a design is A-optimal when $\boldsymbol{H} = \boldsymbol{I}$ in (P-A), whereas the notation V-optimal is reserved for the case when $\boldsymbol{H}$ is arbitrary [17]. We do not make this distinction here for conciseness.

A design is E-optimal if instead of minimizing the MSE as in (P-A), it minimizes the largest eigenvalue of the covariance matrix $\boldsymbol{K}(\mathcal{D})$, i.e.,

$$\underset{|\mathcal{D}| \le k}{\text{minimize}} \quad \lambda_{\max}\left[\boldsymbol{K}(\mathcal{D})\right] - \lambda_{\max}\left[\boldsymbol{H}\boldsymbol{R}_\theta\boldsymbol{H}^T\right]. \tag{P-E}$$

Since the trace of a matrix is the sum of its eigenvalues, we can think of (P-E) as a robust version of (P-A). While the design in (P-A) seeks to reduce the estimation error in all directions, the design in (P-E) seeks to reduce the estimation error in the worst direction. Equivalently, given that $\lambda_{\max}(\boldsymbol{X}) = \max_{\|\boldsymbol{u}\|_2=1} \boldsymbol{u}^T\boldsymbol{X}\boldsymbol{u}$, we can interpret (P-E) with $\boldsymbol{H} = \boldsymbol{I}$ as minimizing the MSE for an adversarial choice of $\boldsymbol{z}$.

A D-optimal design is one in which the objective is to minimize the log-determinant of the estimator's covariance matrix,

$$\underset{|\mathcal{D}| \le k}{\text{minimize}} \quad \log\det\left[\boldsymbol{K}(\mathcal{D})\right] - \log\det\left[\boldsymbol{H}\boldsymbol{R}_\theta\boldsymbol{H}^T\right]. \tag{P-D}$$

The motivation for using the objective in (P-D) is that the log-determinant of $\boldsymbol{K}(\mathcal{D})$ is proportional to the volume of the confidence ellipsoid when the data are Gaussian. Note that the trace, maximum eigenvalue, and determinant of $\boldsymbol{H}\boldsymbol{R}_\theta\boldsymbol{H}^T$ in (P-A), (P-E), and (P-D) are constants and do not affect the respective optimization problems. They are subtracted so that the objectives vanish when $\mathcal{D} = \emptyset$, the empty set. This simplifies the exposition in Section 4.

Although the problem formulations in (P-A), (P-E), and (P-D) are integer programs known to be NP-hard, the use of greedy methods for their solution is widespread with good performance in practice. In the case of D-optimal design, this is justified theoretically because the objective of (P-D) is supermodular, which implies greedy methods are $(1 - e^{-1})$-optimal [2, 11, 12]. The objectives in (P-A) and (P-E), on the other hand, are *not* be supermodular in general [1, 13, 14] and it is not known why their greedy optimization yields good results in practice—conditions for the MSE to be supermodular exist but are restrictive [1]. The goal of this paper is to derive performance guarantees for greedy solutions of A- and E-optimal design problems. We do so by developing different notions of approximate supermodularity to show that A- and E-optimal design problems are not far from supermodular.

**Remark 1.** Besides its intrinsic value as a minimizer of the volume of the confidence ellipsoid, (P-D) is often used as a surrogate for (P-A), when A-optimality (MSE) is considered the appropriate metric. It is important to point out that this is only justified when the problem has some inherent structure that suggests the minimum volume ellipsoid is somewhat symmetric. Otherwise, since the volume of an ellipsoid can be reduced by decreasing the length of a single principal axis, using (P-D) can lead to designs that perform well—in the MSE sense—along a few directions of the parameter space and poorly along all others. Formally, this can be seen by comparing the variation of the log-determinant and trace functions with respect to the eigenvalues of the PSD matrix $\boldsymbol{K}$,

$$\frac{\partial \log\det(\boldsymbol{K})}{\partial \lambda_j(\boldsymbol{K})} = \frac{1}{\lambda_j(\boldsymbol{K})} \qquad \text{and} \qquad \frac{\partial \text{Tr}(\boldsymbol{K})}{\partial \lambda_j(\boldsymbol{K})} = 1.$$

The gradient of the log-determinant is largest in the direction of the smallest eigenvalue of the error covariance matrix. In contrast, the MSE gives equal weight to all directions of the space. The latter therefore yields balanced, whereas the former tends to flatten the confidence ellipsoid unless the problem has a specific structure.

## 3  Approximate supermodularity

Consider a multiset function $f : \mathcal{P}(\mathcal{E}) \rightarrow \mathbb{R}$ for which the value corresponding to an arbitrary multiset $\mathcal{D} \in \mathcal{P}(\mathcal{E})$ is denoted by $f(\mathcal{D})$. We say the function $f$ is normalized if $f(\emptyset) = 0$ and we say $f$ is monotone decreasing if for all multisets $\mathcal{A} \subseteq \mathcal{B}$ it holds that $f(\mathcal{A}) \geq f(\mathcal{B})$. Observe that if a function is normalized and monotone decreasing it must be that $f(\mathcal{D}) \leq 0$ for all $\mathcal{D}$. The objectives of (P-A), (P-E), and (P-D) are normalized and monotone decreasing multiset functions, since adding experiments to a design decreases the covariance matrix uniformly in the PSD cone—see Lemma 1.

We say that a multiset function $f$ is *supermodular* if for all pairs of multisets $\mathcal{A}, \mathcal{B} \in \mathcal{P}(\mathcal{E})$, $\mathcal{A} \subseteq \mathcal{B}$, and elements $u \in \mathcal{E}$ it holds that

$$f(\mathcal{A}) - f(\mathcal{A} \cup \{u\}) \geq f(\mathcal{B}) - f(\mathcal{B} \cup \{u\}).$$

Supermodular functions encode a notion of diminishing returns as the sets grow. Their relevance in this paper is due to the celebrated bound on the suboptimality of their greedy minimization [18]. Specifically, construct a greedy solution by starting with $\mathcal{G}_0 = \emptyset$ and incorporating elements (experiments) $e \in \mathcal{E}$ greedily so that at the $h$-th iteration we incorporate the element whose addition to $\mathcal{G}_{h-1}$ results in the largest reduction in the value of $f$:

$$\mathcal{G}_h = \mathcal{G}_{h-1} \cup \{e\}, \qquad \text{with} \qquad e = \operatorname*{argmin}_{u \in \mathcal{E}} f\left(\mathcal{G}_{h-1} \cup \{u\}\right). \tag{5}$$

The recursion in (5) is repeated for $k$ steps to obtain a greedy solution with $k$ elements. Then, if $f$ is normalized, monotone decreasing, and supermodular,

$$f(\mathcal{G}_k) \leq (1 - e^{-1}) f(\mathcal{D}^\star), \tag{6}$$

where $\mathcal{D}^\star \triangleq \operatorname{argmin}_{|\mathcal{D}| \leq k} f(\mathcal{D})$ is the optimal design selection of cardinality not larger than $k$ [18]. We emphasize that in contrast to the classical greedy algorithm, (5) allows the same element to be selected multiple times.

The optimality guarantee in (6) applies to (P-D) because its objective is supermodular. This is not true of the cost functions of (P-A) and (P-E). We address this issue by postulating that if a function does not violate supermodularity too much, then its greedy minimization should have close to supermodular performance. To formalize this idea, we introduce two measures of approximate supermodularity and derive near-optimal bounds based on these properties. It is worth noting that as intuitive as it may be, such results are not straightforward. In fact, [19] showed that even functions $\delta$-close to supermodular cannot be optimized in polynomial time.

We start with the following multiplicative relaxation of the supermodular property.

**Definition 1** ($\alpha$-supermodularity). *A multiset function* $f : \mathcal{P}(\mathcal{E}) \rightarrow \mathbb{R}$ *is* $\alpha$-supermodular, *for* $\alpha : \mathbb{N} \times \mathbb{N} \rightarrow \mathbb{R}$, *if for all multisets* $\mathcal{A}, \mathcal{B} \in \mathcal{P}(\mathcal{E})$, $\mathcal{A} \subseteq \mathcal{B}$, *and all* $u \in \mathcal{E}$ *it holds that*

$$f\left(\mathcal{A}\right) - f\left(\mathcal{A} \cup \{u\}\right) \geq \alpha(\#\mathcal{A}, \#\mathcal{B})\left[f\left(\mathcal{B}\right) - f\left(\mathcal{B} \cup \{u\}\right)\right]. \tag{7}$$

Notice that for $\alpha \geq 1$, (7) reduces the original definition of supermodularity, in which case we refer to the function simply as supermodular [11, 12]. On the other hand, when $\alpha < 1$, $f$ is said to be approximately supermodular. Notice that if $f$ is decreasing, then (7) always holds for $\alpha \equiv 0$. We are therefore interested in the largest $\alpha$ for which (7) holds, i.e.,

$$\alpha(a, b) = \min_{\substack{\mathcal{A}, \mathcal{B} \in \mathcal{P}(\mathcal{E}) \\ \mathcal{A} \subseteq \mathcal{B}, \ u \in \mathcal{E} \\ \#\mathcal{A} = a, \ \#\mathcal{B} = b}} \frac{f\left(\mathcal{A}\right) - f\left(\mathcal{A} \cup \{u\}\right)}{f\left(\mathcal{B}\right) - f\left(\mathcal{B} \cup \{u\}\right)} \tag{8}$$

Interestingly, $\alpha$ not only measures how much $f$ violates supermodularity, but it also quantifies the loss in performance guarantee incurred from these violations.

**Theorem 1.** *Let $f$ be a normalized, monotone decreasing, and $\alpha$-supermodular multiset function. Then, for $\bar{\alpha} = \min_{a < \ell,\ b < \ell + k} \alpha(a,b)$, the greedy solution from (5) obeys*

$$f(\mathcal{G}_\ell) \leq \left[1 - \prod_{h=0}^{\ell-1}\left(1 - \frac{1}{\sum_{s=0}^{k-1}\alpha(h,h+s)^{-1}}\right)\right]f(\mathcal{D}^\star) \leq (1 - e^{-\bar{\alpha}\ell/k})f(\mathcal{D}^\star). \qquad (9)$$

*Proof.* See extended version [15]. □

Theorem 1 bounds the suboptimality of the greedy solution from (5) when its objective is $\alpha$-supermodular. At the same time, it quantifies the effect of relaxing the supermodularity hypothesis typically used to provide performance guarantees in these settings. In fact, if $f$ is supermodular ($\alpha \equiv 1$) and for $\ell = k$, we recover the $1 - e^{-1} \approx 0.63$ guarantee from [18]. On the other hand, for an approximately supermodular function ($\bar{\alpha} < 1$), the result in (9) shows that the 63% guarantee can be recovered by selecting a set of size $\ell = \bar{\alpha}^{-1}k$. Thus, $\alpha$ not only measures how much $f$ violates supermodularity, but also gives a factor by which the cardinality constraint must be violated to obtain a supermodular near-optimal certificate. Similar to the original bound in [18], it worth noting that (9) is not tight and that better results are typical in practice (see Section 5).

Although $\alpha$-supermodularity gives a multiplicative approximation factor, finding meaningful bounds on $\alpha$ can be challenging for certain multiset functions, such as the E-optimality criterion in (P-E). It is therefore useful to look at approximate supermodularity from a different perspective as in the following definition.

**Definition 2** ($\epsilon$-supermodularity). *A multiset function $f : \mathcal{P}(\mathcal{E}) \to \mathbb{R}$ is $\epsilon$-supermodular, for $\epsilon : \mathbb{N} \times \mathbb{N} \to \mathbb{R}$, if for all multisets $\mathcal{A}, \mathcal{B} \in \mathcal{P}(\mathcal{E})$, $\mathcal{A} \subseteq \mathcal{B}$, and all $u \in \mathcal{E}$ it holds that*

$$f(\mathcal{A}) - f(\mathcal{A} \cup \{u\}) \geq f(\mathcal{B}) - f(\mathcal{B} \cup \{u\}) - \epsilon(\#\mathcal{A}, \#\mathcal{B}). \qquad (10)$$

Again, we say $f$ is supermodular if $\epsilon(a,b) \leq 0$ for all $a, b$ and approximately supermodular otherwise. As with $\alpha$, we want the best $\epsilon$ that satisfies (10), which is given by

$$\epsilon(a,b) = \max_{\substack{\mathcal{A},\mathcal{B} \in \mathcal{P}(\mathcal{E}) \\ \mathcal{A} \subseteq \mathcal{B},\ u \in \mathcal{E} \\ \#\mathcal{A}=a,\ \#\mathcal{B}=b}} f(\mathcal{B}) - f(\mathcal{B} \cup \{u\}) - f(\mathcal{A}) + f(\mathcal{A} \cup \{u\}). \qquad (11)$$

In contrast to $\alpha$-supermodularity, we obtain an additive approximation guarantee for the greedy minimization of $\epsilon$-supermodular functions.

**Theorem 2.** *Let $f$ be a normalized, monotone decreasing, and $\epsilon$-supermodular multiset function. Then, for $\bar{\epsilon} = \max_{a < \ell,\ b < \ell + k} \epsilon(a,b)$, the greedy solution from (5) obeys*

$$f(\mathcal{G}_\ell) \leq \left[1 - \left(1 - \frac{1}{k}\right)^\ell\right]f(\mathcal{D}^\star) + \frac{1}{k}\sum_{s=0}^{k-1}\sum_{h=0}^{\ell-1}\epsilon(h,h+s)\left(1 - \frac{1}{k}\right)^{\ell-1-h}$$

$$\leq (1 - e^{-\ell/k})(f(\mathcal{D}^\star) + k\bar{\epsilon}) \qquad (12)$$

*Proof.* See extended version [15]. □

As before, $\epsilon$ quantifies the loss in performance guarantee due to relaxing supermodularity. Indeed, (12) reveals that $\epsilon$-supermodular functions have the same guarantees as a supermodular function up to an additive factor of $\Theta(k\bar{\epsilon})$. In fact, if $\bar{\epsilon} \leq (ek)^{-1}|f(\mathcal{D}^\star)|$ (recall that $f(\mathcal{D}^\star) \leq 0$ due to normalization), then taking $\ell = 3k$ recovers the 63% approximation factor of supermodular functions. This same factor is obtained for $\alpha \geq 1/3$-supermodular functions.

With the certificates of Theorems 1 and 2 in hand, we now proceed with the study of the A- and E-optimality criteria. In the next section, we derive explicit bounds on their $\alpha$- and $\epsilon$-supermodularity, respectively, thus providing near-optimal performance guarantees for greedy A- and E-optimal designs.

# 4  Near-optimal experimental design

Theorems 1 and 2 apply to functions that are (i) normalized, (ii) monotone decreasing, and (iii) approximately supermodular. By construction, the objectives of (P-A) and (P-E) are normalized [(i)]. The following lemma establishes that they are also monotone decreasing [(ii)] by showing that $\boldsymbol{K}$ is a decreasing set function in the PSD cone. The definition of Loewner order and the monotonicity of the trace operator readily give the desired results [16].

**Lemma 1.** *The matrix-valued set function $\boldsymbol{K}(\mathcal{D})$ in* (4) *is monotonically decreasing with respect to the PSD cone, i.e., $\mathcal{A} \subseteq \mathcal{B} \Leftrightarrow \boldsymbol{K}(\mathcal{A}) \succeq \boldsymbol{K}(\mathcal{B})$.*

*Proof.* See extended version [15]. □

The main results of this section provide the final ingredient [(iii)] for Theorems 1 and 2 by bounding the approximate supermodularity of the A- and E-optimality criteria. We start by showing that the objective of (P-A) is $\alpha$-supermodular.

**Theorem 3.** *The objective of* (P-A) *is $\alpha$-supermodular with*

$$\alpha(a, b) \geq \frac{1}{\kappa(\boldsymbol{H})^2} \cdot \frac{\lambda_{\min}\left[\boldsymbol{R}_\theta^{-1}\right]}{\lambda_{\max}\left[\boldsymbol{R}_\theta^{-1}\right] + a \cdot \ell_{\max}}, \quad \text{for all } b \in \mathbb{N}, \tag{13}$$

*where $\ell_{\max} = \max_{e \in \mathcal{E}} \lambda_{\max}(\boldsymbol{M}_e)$, $\boldsymbol{M}_e = \boldsymbol{A}_e^T \boldsymbol{R}_e^{-1} \boldsymbol{A}_e$, and $\kappa(\boldsymbol{H}) = \sigma_{\max} / \sigma_{\min}$ is the $\ell_2$-norm condition number of $\boldsymbol{H}$, with $\sigma_{\max}$ and $\sigma_{\min}$ denoting the largest and smallest singular values of $\boldsymbol{H}$ respectively.*

*Proof.* See extended version [15]. □

Theorem 3 bounds the $\alpha$-supermodularity of the objective of (P-A) in terms of the condition number of $\boldsymbol{H}$, the prior covariance matrix, and the measurements SNR. To facilitate the interpretation of this result, let the SNR of the $e$-th experiment be $\gamma_e = \text{Tr}[\boldsymbol{M}_e]$ and suppose $\boldsymbol{R}_\theta = \sigma_\theta^2 \boldsymbol{I}$, $\boldsymbol{H} = \boldsymbol{I}$, and $\gamma_e \leq \gamma$ for all $e \in \mathcal{E}$. Then, for $\ell = k$ greedy iterations, (13) implies

$$\bar{\alpha} \geq \frac{1}{1 + 2k\sigma_\theta^2 \gamma},$$

for $\bar{\alpha}$ as in Theorem 1. This deceptively simple bound reveals that the MSE behaves as a supermodular function at low SNRs. Formally, $\alpha \to 1$ as $\gamma \to 0$. In contrast, the performance guarantee from Theorem 3 degrades in high SNR scenarios. In this case, however, greedy methods are expected to give good results since designs yield similar estimation errors (as illustrated in Section 5). The greedy solution of (P-A) also approaches the $1 - 1/e$ guarantee when the prior on $\boldsymbol{\theta}$ is concentrated ($\sigma_\theta^2 \ll 1$), i.e., when the problem is heavily regularized.

These observations also hold for a generic $\boldsymbol{H}$ as long as it is well-conditioned. Even if $\kappa(\boldsymbol{H}) \gg 1$, we can replace $\boldsymbol{H}$ by $\tilde{\boldsymbol{H}} = \boldsymbol{D}\boldsymbol{H}$ for some diagonal matrix $\boldsymbol{D} \succ 0$ without affecting the design, since $\boldsymbol{z}$ is arbitrarily scaled. The scaling $\boldsymbol{D}$ can be designed to minimize the condition number of $\tilde{\boldsymbol{H}}$ by leveraging preconditioning and balancing methods [20, 21].

Proceeding, we derive guarantees for E-optimal designs using $\epsilon$-supermodularity.

**Theorem 4.** *The cost function of* (P-E) *is $\epsilon$-supermodular with*

$$\epsilon(a, b) \leq (b - a)\, \sigma_{\max}(\boldsymbol{H})^2\, \lambda_{\max}\left(\boldsymbol{R}_\theta\right)^2 \ell_{max}, \tag{14}$$

*where $\ell_{max} = \max_{e \in \mathcal{E}} \lambda_{\max}(\boldsymbol{M}_e)$, $\boldsymbol{M}_e = \boldsymbol{A}_e^T \boldsymbol{R}_e^{-1} \boldsymbol{A}_e$, and $\sigma_{\max}(\boldsymbol{H})$ is the largest singular value of $\boldsymbol{H}$.*

*Proof.* See extended version [15]. □

Under the same assumptions as above, Theorem 4 gives

$$\bar{\epsilon} \leq 2k\sigma_\theta^4 \gamma,$$

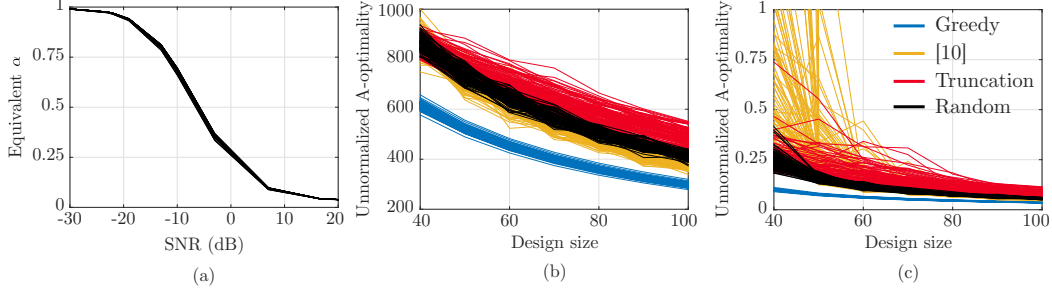

Figure 1: A-optimal design: (a) Thm. 3; (b) A-optimality (low SNR); (c) A-optimality (high SNR). The plots show the unnormalized A-optimality value for clarity.

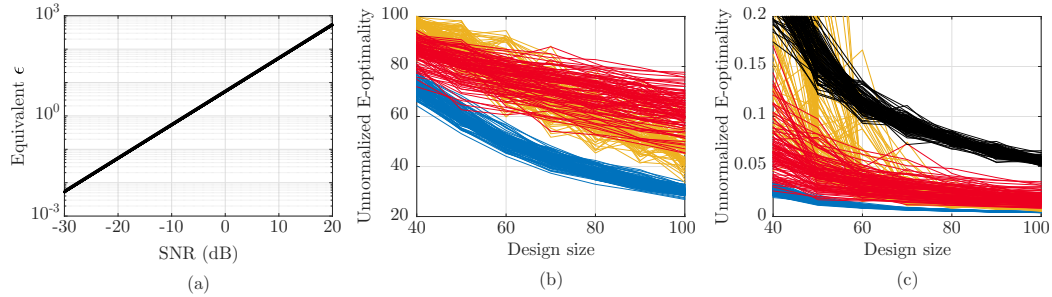

Figure 2: E-optimal design: (a) Thm. 4; (b) E-optimality (low SNR); (c) E-optimality (high SNR). The plots show the unnormalized E-optimality value for clarity.

for $\bar{\epsilon}$ as in Theorem 2. Thus, $\epsilon \to 0$ as $\gamma \to 0$. In other words, the behavior of the objective of (P-E) approaches that of a supermodular function as the SNR decreases. The same holds for concentrated priors, i.e., $\lim_{\sigma_\theta^2 \to 0} \bar{\epsilon} = 0$. Once again, it is worth noting that when the SNRs of the experiments are large, almost every design has the same E-optimal performance as long as the experiments are not too correlated. Thus, greedy design is also expected to give good results under these conditions.

Finally, the proofs of Theorems 3 and 4 suggest that better bounds can be found when the designs are constructed without replacement, i.e., when only one of each experiment is allowed in the design.

## 5 Numerical examples

In this section, we illustrate the previous results in some numerical examples. To do so, we draw the elements of $\boldsymbol{A}_e$ from an i.i.d. zero-mean Gaussian random variable with variance $1/p$ and $p = 20$. The noise $\{\boldsymbol{v}_e\}$ are also Gaussian random variables with $\boldsymbol{R}_e = \sigma_v^2 \boldsymbol{I}$. We take $\sigma_v^2 = 10^{-1}$ in high SNR and $\sigma_v^2 = 10$ in low SNR simulations. The experiment pool contains $\#\mathcal{E} = 200$ experiments.

Starting with A-optimal design, we display the bound from Theorem 3 in Figure 1a for multivariate measurements of size $n_e = 5$ and designs of size $k = 40$. Here, "equivalent $\alpha$" is the single $\hat{\alpha}$ that gives the same near-optimal certificate (9) as using (13). As expected, $\hat{\alpha}$ approaches 1 as the SNR decreases. In fact, for $-10$ dB is is already close to $0.75$ which means that by selecting a design of size $\ell = 55$ we would be within $1 - 1/e$ of the optimal design of size $k = 40$. Figures 1b and 1c compare greedy A-optimal designs with the convex relaxation of (P-A) in low and high SNR scenarios. The designs are obtained from the continuous solutions using the hard constraint, with replacement method of [10] and a simple design truncation as in [22]. Therefore, these simulations consider univariate measurements ($n_e = 1$). For comparison, a design sampled uniformly at random with replacement from $\mathcal{E}$ is also presented. Note that, as mentioned before, the performance difference across designs is small for high SNR—notice the scale in Figures 1c and 2c—, so that even random designs perform well.

For the E-optimality criterion, the bound from Theorem 4 is shown in Figure 2a, again for multivariate measurements of size $n_e = 5$ and designs of size $k = 40$. Once again, "equivalent $\epsilon$" is the single value $\hat{\epsilon}$ that yields the same guarantee as using (14). In this case, the bound degradation in

high SNR is more pronounced. This reflects the difficulty in bounding the approximate supermodularity of the E-optimality cost function. Still, Figures 2b and 2c show that greedy E-optimal designs have good performance when compared to convex relaxations or random designs. Note that, though it is not intended for E-optimal designs, we again display the results of the sampling post-processing from [10]. In Figure 2b, the random design is omitted due to its poor performance.

## 5.1 Cold-start survey design for recommender systems

Recommender systems use semi-supervised learning methods to predict user ratings based on few rated examples. These methods are useful, for instance, to streaming service providers who are interested in using predicted ratings of movies to provide recommendations. For new users, these systems suffer from a "cold-start problem," which refers to the fact that it is hard to provide accurate recommendations without knowing a user's preference on at least a few items. For this reason, services explicitly ask users for ratings in initial surveys before emitting any recommendation. Selecting which movies should be rated to better predict a user's preferences can be seen as an experimental design problem. In the following example, we use a subset of the EachMovie dataset [23] to illustrate how greedy experimental design can be applied to address this problem.

We randomly selected a training and test set containing 9000 and 3000 users respectively. Following the notation from Section 2, each experiment in $\mathcal{E}$ represents a movie ($|\mathcal{E}| = 1622$) and the observation vector $\boldsymbol{A}_e$ collects the ratings of movie $e$ for each user in the training set. The parameter $\boldsymbol{\theta}$ is used to express the rating of a new user in term of those in the training set. Our hope is that we can extrapolate the observed ratings, i.e., $\{y_e\}_{e \in \mathcal{D}}$, to obtain the rating for a movie $f \notin \mathcal{D}$ as $\hat{y}_f = \boldsymbol{A}_f \hat{\boldsymbol{\theta}}$. Since the mean absolute error (MAE) is commonly used in this setting, we choose to work with the A-optimality criterion. We also let $\boldsymbol{H} = \boldsymbol{I}$ and take a non-informative prior $\bar{\boldsymbol{\theta}} = \boldsymbol{0}$ and $\boldsymbol{R}_\theta = \sigma_\theta^2 \boldsymbol{I}$ with $\sigma_\theta^2 = 100$.

As expected, greedy A-optimal design is able to find small sets of movies that lead to good prediction. For $k = 10$, for example, MAE $= 2.3$, steadily reducing until MAE $< 1.8$ for $k \geq 35$. These are considerably better results than a random movie selection, for which the MAE varies between $2.8$ and $3.3$ for $k$ between $10$ and $50$. Instead of focusing on the raw ratings, we may be interested in predicting the user's favorite genre. This is a challenging task due to the heavily skewed dataset. For instance, $32\%$ of the movies are dramas whereas only $0.02\%$ are animations. Still, we use the simplest possible classifier by selecting the category with highest average estimated ratings. By using greedy design, we can obtain a misclassification rate of approximately $25\%$ by observing $100$ ratings, compared to over $45\%$ error rate for a random design.

# 6 Related work

**Optimal experimental design**   Classical experimental design typically relies on convex relaxations to solve optimal design problems [17, 22]. However, because these are semidefinite programs (SDPs) or sequential second-order cone programs (SOCPs), their computational complexity can hinder their use in large-scale problems [5, 7, 22, 24]. Another issue with these relaxations is that some sort of post-processing is required to extract a valid design from their continuous solutions [5, 22]. For D-optimal designs, this can be done with $(1 - e^{-1})$-optimality [25, 26]. For A-optimal designs, [10] provides near-optimal randomized schemes for large enough $k$.

**Greedy optimization guarantees**   The $(1 - e^{-1})$-suboptimality of greedy search for supermodular minimization under cardinality constraints was established in [18]. To deal with the fact that the MSE is not supermodular, $\alpha$-supermodularity with constant $\alpha$ was introduced in [27] along with explicit lower bounds. This concept is related to the *submodularity ratio* introduced by [3] to obtain guarantees similar to Theorem 1 for dictionary selection and forward regression. However, the bounds on the submodularity ratio from [3, 28] depend on the sparse eigenvalues of $\boldsymbol{K}$ or restricted strong convexity constants of the A-optimal objective, which are NP-hard to compute. Explicit bounds for the submodularity ratio of A-optimal experimental design were recently obtained in [29]. Nevertheless, neither [27] nor [29] consider multisets. Hence, to apply their results we must operate on an extended ground set containing $k$ unique copies of each experiment, which make the bounds uninformative. For instance, in the setting of Section 5, Theorem 3 guarantees $0.1$-optimality at $0$ dB SNR whereas [29] guarantees $2.5 \times 10^{-6}$-optimality. The concept of $\epsilon$-supermodularity was

first explored in [30] for a constant $\epsilon$. There, guarantees for dictionary selection were derived by bounding $\epsilon$ using an incoherence assumption on the $\boldsymbol{A}_e$. Finally, a more stringent definition of approximately submodular functions was put forward in [19] by requiring the function to be upper and lower bounded by a submodular function. They show strong impossibility results unless the function is $\mathcal{O}(1/k)$-close to submodular. *Approximate submodularity* is sometimes referred to as *weak submodularity* (e.g., [28]), though it is not related to the weak submodularity concept from [31].

## 7 Conclusions

Greedy search is known to be an empirically effective method to find A- and E-optimal experimental designs despite the fact that these objectives are not supermodular. We reconciled these observations by showing that the A- and E-optimality criteria are approximately supermodular and deriving near-optimal guarantees for this class of functions. By quantifying their supermodularity violations, we showed that the behavior of the MSE and the maximum eigenvalue of the error covariance matrix becomes increasingly supermodular as the SNR decreases. An important open question is whether these results can be improved using additional knowledge. Can we exploit some structure of the observation matrices (e.g., Fourier, random)? What if the parameter vector is sparse but with unknown support (e.g., compressive sensing)? Are there practical experiment properties other than the SNR that lead to small supermodular violations? Finally, we hope that this approximate supermodularity framework can be extended to other problems.

### Acknowledgments

This work was supported by the National Science Foundation CCF 1717120 and in part by the ARO W911NF1710438.

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
