[Supplementary Material · nips2017_supplementary.pdf]

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

(H) \gg 1$, we can replace $H$ by $\tilde{H} = DH$ for some diagonal matrix $D \succ 0$ without affecting the design, since $z$ is arbitrarily scaled. The scaling $D$ can be designed to minimize the condition number of $\tilde{H}$ by leveraging preconditioning and balancing methods [20, 21].

Proceeding, we derive guarantees for E-optimal designs using $\epsilon$-supermodularity.

**Theorem 4.** *The cost function of* (P-E) *is $\epsilon$-supermodular with*

$$\epsilon(a,b) \leq (b-a)\, \sigma_{\max}(H)^2\, \lambda_{\max}\left(R_\theta\right)^2 \ell_{max}, \tag{14}$$

*where $\ell_{max} = \max_{e \in \mathcal{E}} \lambda_{\max}(M_e)$, $M_e = A_e^T R_e^{-1} A_e$, and $\sigma_{\max}(H)$ is the largest singular value of $

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

# A  Proofs

*Proof of Proposition 1.* Start by lifting the problem to make the proof more concise by defining the stacked quantities $\tilde{\boldsymbol{y}}_{\mathcal{D}} = [\boldsymbol{y}_e]_{e \in \mathcal{D}}$, an $n \times 1$ vector, $\tilde{\boldsymbol{A}}_{\mathcal{D}} = [\boldsymbol{A}_e]_{e \in \mathcal{D}}$, an $n \times p$ matrix, $\tilde{\boldsymbol{v}}_{\mathcal{D}} = [\boldsymbol{v}_e]_{e \in \mathcal{D}}$, an $n \times 1$ vector, and $\tilde{\boldsymbol{R}}_{\mathcal{D}} = \mathrm{blkdiag}(\boldsymbol{R}_e)_{e \in \mathcal{D}}$, an $n \times n$ block diagonal matrix, with $n = \sum_{j \in \mathcal{D}} n_j$. Since the design is fixed, the dependence on $\mathcal{D}$ is omitted throughout the proof for clarity.

Note that all affine estimators of $\boldsymbol{z}$ can be written as $\hat{\boldsymbol{z}} = \boldsymbol{L}\tilde{\boldsymbol{y}} + \boldsymbol{b}$, so that the problem reduces to determining the optimal $\boldsymbol{L}^\star$ and $\boldsymbol{b}^\star$. Using the model in (1), write $\boldsymbol{L}\tilde{\boldsymbol{y}} = \boldsymbol{L}\left(\tilde{\boldsymbol{A}}\boldsymbol{\theta} + \tilde{\boldsymbol{v}}\right)$, so that the error covariance matrix of the estimator has the form

$$\boldsymbol{K} = \mathbb{E}\left[(\boldsymbol{H}\boldsymbol{\theta} - \boldsymbol{L}\tilde{\boldsymbol{A}}\boldsymbol{\theta} - \boldsymbol{L}\tilde{\boldsymbol{v}} - \boldsymbol{b})(\boldsymbol{H}\boldsymbol{\theta} - \boldsymbol{L}\tilde{\boldsymbol{A}}\boldsymbol{\theta} - \boldsymbol{L}\tilde{\boldsymbol{v}} - \boldsymbol{b})^T \mid \boldsymbol{\theta}, \tilde{\boldsymbol{R}}\right]$$
$$= (\boldsymbol{H} - \boldsymbol{L}\tilde{\boldsymbol{A}})\boldsymbol{R}_{\theta}(\boldsymbol{H} - \boldsymbol{L}\tilde{\boldsymbol{A}})^T + \boldsymbol{L}\tilde{\boldsymbol{R}}\boldsymbol{L}^T$$
$$+ \left[(\boldsymbol{H} - \boldsymbol{L}\tilde{\boldsymbol{A}})\bar{\boldsymbol{\theta}} - \boldsymbol{b}\right]\left[(\boldsymbol{H} - \boldsymbol{L}\tilde{\boldsymbol{A}})\bar{\boldsymbol{\theta}} - \boldsymbol{b}\right]^T,$$

where all terms linear in $\bar{\boldsymbol{v}}$ vanish since $\{\boldsymbol{v}_e, \boldsymbol{\theta}\}$ are independent for all $e \in \mathcal{E}$. It is ready that the last term is minimized by taking
$$\boldsymbol{b}^\star = (\boldsymbol{H} - \boldsymbol{L}\tilde{\boldsymbol{A}})\bar{\boldsymbol{\theta}}.$$
Suffices now to minimize the sum of the first two terms.

To do so, note that for $\boldsymbol{b} = \boldsymbol{b}^\star$, we have $\boldsymbol{K} = (\boldsymbol{H} - \boldsymbol{L}\tilde{\boldsymbol{A}})\boldsymbol{R}_{\theta}(\boldsymbol{H} - \boldsymbol{L}\tilde{\boldsymbol{A}})^T$. Therefore, taking $\boldsymbol{L} = \boldsymbol{L}^\star + (\boldsymbol{L} - \boldsymbol{L}^\star)$ with

$$\boldsymbol{L}^\star = \boldsymbol{H}\left(\boldsymbol{R}_{\theta}^{-1} + \tilde{\boldsymbol{A}}^T\tilde{\boldsymbol{R}}^{-1}\tilde{\boldsymbol{A}}\right)^{-1}\tilde{\boldsymbol{A}}^T\tilde{\boldsymbol{R}}^{-1},$$

and expanding gives

$$\boldsymbol{K} = \left(\boldsymbol{H} - \boldsymbol{L}^\star\tilde{\boldsymbol{A}}\right)\boldsymbol{R}_{\theta}\left(\boldsymbol{H} - \boldsymbol{L}^\star\tilde{\boldsymbol{A}}\right)^T + \boldsymbol{L}^\star\tilde{\boldsymbol{R}}\boldsymbol{L}^{\star T}$$
$$+ (\boldsymbol{L} - \boldsymbol{L}^\star)\left(\tilde{\boldsymbol{A}}\boldsymbol{R}_{\theta}\tilde{\boldsymbol{A}}^T + \tilde{\boldsymbol{R}}\right)(\boldsymbol{L} - \boldsymbol{L}^\star)^T$$
$$+ (\boldsymbol{L} - \boldsymbol{L}^\star)\left[\boldsymbol{L}^\star\bar{\boldsymbol{R}} - \left(\boldsymbol{H} - \boldsymbol{L}^\star\tilde{\boldsymbol{A}}\right)\boldsymbol{R}_{\theta}\tilde{\boldsymbol{A}}^T\right]^T$$
$$+ \left[\boldsymbol{L}^\star\tilde{\boldsymbol{R}} - \left(\boldsymbol{H} - \boldsymbol{L}^\star\tilde{\boldsymbol{A}}\right)\boldsymbol{R}_{\theta}\tilde{\boldsymbol{A}}^T\right](\boldsymbol{L} - \boldsymbol{L}^\star)^T$$
$$= \boldsymbol{K}^\star + (\boldsymbol{L} - \boldsymbol{L}^\star)\left(\tilde{\boldsymbol{A}}\boldsymbol{R}_{\theta}\tilde{\boldsymbol{A}}^T + \tilde{\boldsymbol{R}}\right)(\boldsymbol{L} - \boldsymbol{L}^\star)^T, \tag{15}$$

where the two last terms vanish and $\boldsymbol{K}^\star = (\boldsymbol{H} - \boldsymbol{L}^\star\tilde{\boldsymbol{A}})\boldsymbol{R}_{\theta}(\boldsymbol{H} - \boldsymbol{L}^\star\tilde{\boldsymbol{A}})^T + \boldsymbol{L}^\star\tilde{\boldsymbol{R}}\boldsymbol{L}^{\star T}$. Clearly, the minimum value of (15) is $\boldsymbol{K}^\star$, attained for $\boldsymbol{L} = \boldsymbol{L}^\star$. Finally, $\hat{\boldsymbol{z}} = \boldsymbol{L}^\star\tilde{\boldsymbol{y}} + \boldsymbol{b}^\star$ and $\boldsymbol{K}^\star$ can be unstacked and rearranged to yield (3) and (4). $\qquad\square$

*Proof of Theorem 1.* Since $f$ is monotone decreasing, it holds for every $\mathcal{G}_h$ that

$$f(\mathcal{D}^\star) \geq f(\mathcal{D}^\star \cup \mathcal{G}_h) = f(\mathcal{G}_h) + \sum_{s=0}^{k-1} f(\mathcal{T}_s \cup \{e_s^\star\}) - f(\mathcal{T}_s), \tag{16}$$

where $\mathcal{T}_s = \mathcal{G}_h \cup \{e_0^\star, \dots, e_{s-1}^\star\}$, with $\mathcal{T}_0 = \mathcal{G}_h$, and $e_s^\star$ is the $s$-th experiment in $\mathcal{D}^\star$. The equality comes from expressing the set function as a telescopic sum. Since $f$ is $\alpha$-supermodular and $\mathcal{G}_h \subseteq \mathcal{T}_s$ for all $s$, the incremental gains in (16) can be bounded using (7) to get

$$f(\mathcal{D}^\star) \geq f(\mathcal{G}_h) + \sum_{s=0}^{k-1} \alpha(h, h+s)^{-1}\left[f(\mathcal{G}_h \cup \{e_s^\star\}) - f(\mathcal{G}_h)\right].$$

Given that $\mathcal{G}_{h+1}$ is construct from $\mathcal{G}_h$ so as to minimize $f(\mathcal{G}_{h+1})$ [as in (5)], it holds that

$$f(\mathcal{D}^\star) \geq f(\mathcal{G}_h) + [f(\mathcal{G}_{h+1}) - f(\mathcal{G}_h)]\sum_{s=0}^{k-1} \alpha(h, h+s)^{-1}. \tag{17}$$

A recursion is obtained by taking $\delta_h = f(\mathcal{G}_h) - f(\mathcal{D}^\star)$, so that (17) can be written as

$$\delta_h \le \alpha_t(h)\,(\delta_h - \delta_{h+1}) \Rightarrow \delta_{h+1} \le \left(1 - \frac{1}{\alpha_t(h)}\right)\delta_h,$$

with $\alpha_t(h) = \sum_{s=0}^{k-1} \alpha(h, h+s)^{-1}$. Considering that $f$ is normalized, $\delta_0 = -f(\mathcal{D}^\star)$ and the solution of this recursion is

$$f(\mathcal{G}_\ell) \le \left[1 - \prod_{h=0}^{\ell-1}\left(1 - \frac{1}{\alpha_t(h)}\right)\right]f(\mathcal{D}^\star).$$

Since $\bar{\alpha} \le \alpha(h, h+s)$ for $h < \ell$ and $s < k$, it holds that $\alpha_t(h) \le \bar{\alpha}^{-1}k$. Then, using the fact that $1 - x \le e^{-x}$ yields (9). $\qquad\square$

*Proof of Theorem 2.* Given that $f$ is monotone decreasing,

$$f(\mathcal{D}^\star) \ge f(\mathcal{D}^\star \cup \mathcal{G}_h) = f(\mathcal{G}_h) + \sum_{s=0}^{k-1} f(\mathcal{T}_s \cup \{e_s^\star\}) - f(\mathcal{T}_s), \qquad (18)$$

where $\mathcal{T}_s = \mathcal{G}_h \cup \{e_0^\star, \ldots, e_{s-1}^\star\}$, with $\mathcal{T}_0 = \mathcal{G}_h$, and $e_s^\star$ is the $s$-th experiment in $\mathcal{D}^\star$. Since $f$ is $\epsilon$-supermodular and $\mathcal{G}_h \subseteq \mathcal{T}_s$ for all $s$, (10) can be used to bound the incremental gains in (18). Explicitly,

$$f(\mathcal{D}^\star) \ge f(\mathcal{G}_h) + \sum_{s=0}^{k-1} \left[f(\mathcal{G}_h \cup \{e_s^\star\}) - f(\mathcal{G}_h) + \epsilon(h, h+s)\right].$$

Since $\mathcal{G}_{h+1}$ is chosen greedily to minimize $f(\mathcal{G}_{h+1})$ [see (5)], it holds that

$$f(\mathcal{D}^\star) \ge f(\mathcal{G}_h) + k\left[f(\mathcal{G}_{h+1}) - f(\mathcal{G}_h)\right] + \sum_{s=0}^{k-1} \epsilon(h, h+s). \qquad (19)$$

The following recursion is obtained from (19) by letting $\delta_h = f(\mathcal{G}_h) - f(\mathcal{D}^\star)$:

$$\delta_h \le k\,(\delta_h - \delta_{h+1}) - \epsilon_t(h) \Rightarrow \delta_{h+1} \le \left(1 - \frac{1}{k}\right)\delta_h + \frac{\epsilon_t(h)}{k}$$

with $\epsilon_t(h) = \sum_{s=0}^{k-1} \epsilon(h, h+s)$. Since $f$ is normalized, $\delta_0 = -f(\mathcal{D}^\star)$, and solving this recursion yields

$$f(\mathcal{G}_\ell) \le \left[1 - \left(1 - \frac{1}{k}\right)^\ell\right]f(\mathcal{D}^\star) + \frac{1}{k}\sum_{h=0}^{\ell-1}\epsilon_t(h)\left(1 - \frac{1}{k}\right)^{\ell-1-h}.$$

Using the fact that $\bar{\epsilon} \ge \epsilon(h, h+s)$ for $h < \ell$ and $s < k$ then gives

$$f(\mathcal{G}_\ell) \le \left[1 - \left(1 - \frac{1}{k}\right)^\ell\right]f(\mathcal{D}^\star) + \bar{\epsilon}\sum_{h=0}^{\ell-1}\left(1 - \frac{1}{k}\right)^h,$$

from which (12) obtains using the closed form of the geometric series and $1 - x \le e^{-x}$. $\qquad\square$

*Proof of Lemma 1.* Start by noting that $\boldsymbol{K}$ in (4) can be written as

$$\boldsymbol{K}(\mathcal{D}) = \boldsymbol{H}\bar{\boldsymbol{K}}(\mathcal{D})\boldsymbol{H}^T,$$

with $\bar{\boldsymbol{K}}(\mathcal{D}) = \boldsymbol{Y}(\mathcal{D})^{-1}$ and $\boldsymbol{Y}(\mathcal{D}) = \boldsymbol{R}_\theta^{-1} + \sum_{e \in \mathcal{D}} \boldsymbol{M}_e$. Since $\boldsymbol{K}$ and $\bar{\boldsymbol{K}}$ are congruent, suffices to show that $\boldsymbol{K}$ is a monotonically decreasing (with respect to the PSD cone) set function [16].

To do so, note that $\boldsymbol{Y}(\mathcal{A} \cup \mathcal{B}) = \boldsymbol{Y}(\mathcal{A}) + \sum_{e \in \mathcal{B} \setminus \mathcal{A}} \boldsymbol{M}_e$. Then, since $\boldsymbol{M}_e = \boldsymbol{A}_e^T \boldsymbol{R}_e^{-1} \boldsymbol{A}_e$ and $\boldsymbol{R}_e \succ 0$, $\boldsymbol{Y}$ is a sum of PSD matrices, which implies that $\mathcal{A} \subseteq \mathcal{B} \Rightarrow \boldsymbol{Y}(\mathcal{A}) \preceq \boldsymbol{Y}(\mathcal{B})$, i.e., $\boldsymbol{Y}$ is monotonically increasing. From the antitonicity of the matrix inverse [32], it follows that $\bar{\boldsymbol{K}}$ is monotonically decreasing. $\qquad\square$

*Proof of Theorem 3.* This proof relies on the fact that $\alpha$ depends only on rank-one updates of the covariance matrix. Therefore, we can use the matrix inversion lemma to obtain a closed-form expression for the increments required to evaluate (8). Spectral inequalities are then used to bound the increments ratio.

Explicitly, start by expressing the error covariance matrix from (4) as $\boldsymbol{K}(\mathcal{D}) = \boldsymbol{H}\boldsymbol{Y}(\mathcal{D})^{-1}\boldsymbol{H}^T$, with $\boldsymbol{Y}(\mathcal{D}) = \boldsymbol{R}_\theta^{-1} + \sum_{e\in\mathcal{D}}\boldsymbol{M}_e$ and define $g(\mathcal{D}) = \mathrm{Tr}[\boldsymbol{K}(\mathcal{D})]$. Then, notice that $\alpha$ only depends on the incremental gains $\Delta_u(\mathcal{X}) = g(\mathcal{X}) - g(\mathcal{X}\cup\{u\})$. Indeed, (8) can be rewritten as

$$\alpha(a,b) = \min_{\substack{\mathcal{A},\mathcal{B}\in\mathcal{P}(\mathcal{E}) \\ \mathcal{A}\subseteq\mathcal{B},\ u\in\mathcal{E} \\ \#\mathcal{A}=a,\ \#\mathcal{B}=b}} \frac{\Delta_u(\mathcal{A})}{\Delta_u(\mathcal{B})}. \tag{20}$$

Using the additivity of $\boldsymbol{Y}$ gives $g(\mathcal{X}\cup\{u\}) = \mathrm{Tr}\left[\boldsymbol{H}\left(\boldsymbol{Y}(\mathcal{X})+\boldsymbol{M}_u\right)^{-1}\boldsymbol{H}^T\right]$, which suggests that the matrix inversion lemma can be used to obtain a simpler expression for $\Delta$. However, although $\boldsymbol{Y}(\mathcal{X}) \succ 0$ due to $\boldsymbol{R}_\theta \succ 0$, the matrices $\boldsymbol{M}_u$ need not be invertible. Thus, we use the inversion lemma version from [33] to get

$$g(\mathcal{X}\cup\{u\}) = \mathrm{Tr}\left[\boldsymbol{H}\boldsymbol{Y}(\mathcal{X})^{-1}\boldsymbol{H}^T - \boldsymbol{H}\boldsymbol{Y}(\mathcal{X})^{-1}\boldsymbol{M}_u\left[\boldsymbol{Y}(\mathcal{X})+\boldsymbol{M}_u\right]^{-1}\boldsymbol{H}^T\right].$$

Finally, the linearity of the trace operator implies

$$\Delta_u(\mathcal{X}) = \mathrm{Tr}\left[\boldsymbol{H}\boldsymbol{Y}(\mathcal{X})^{-1}\boldsymbol{M}_u\left[\boldsymbol{Y}(\mathcal{X})+\boldsymbol{M}_u\right]^{-1}\boldsymbol{H}^T\right]. \tag{21}$$

Our goal is now to explicitly lower bound (20) by exploiting the expression in (21) and spectral bounds. We do so by using the following result:

**Lemma 2.** *For all $\mathcal{X}\subseteq\mathcal{P}(\mathcal{E})$ and $u\in\mathcal{E}$, it holds that for $\Delta$ as in (21)*

$$\lambda_{\min}\left[\boldsymbol{H}\boldsymbol{H}^T\right]\lambda_{\min}\left[\boldsymbol{Y}(\mathcal{X})^{-1}\right]\mathrm{Tr}\left[\boldsymbol{M}_u\left[\boldsymbol{Y}(\mathcal{X})+\boldsymbol{M}_u\right]^{-1}\right] \leq \Delta_u(\mathcal{X}) \leq$$
$$\leq \lambda_{\max}\left[\boldsymbol{H}\boldsymbol{H}^T\right]\lambda_{\max}\left[\boldsymbol{Y}(\mathcal{X})^{-1}\right]\mathrm{Tr}\left[\boldsymbol{M}_u\left[\boldsymbol{Y}(\mathcal{X})+\boldsymbol{M}_u\right]^{-1}\right]. \tag{22}$$

Before proving Lemma 2, let us see how it leads to the desired result. Using (22) we can bound (20) as in

$$\alpha(a,b) \geq \min_{\substack{\mathcal{A},\mathcal{B}\in\mathcal{P}(\mathcal{E}) \\ \mathcal{A}\subseteq\mathcal{B},\ u\in\mathcal{E} \\ \#\mathcal{A}=a,\ \#\mathcal{B}=b}} \frac{\lambda_{\min}\left(\boldsymbol{H}\boldsymbol{H}^T\right)\lambda_{\min}\left[\boldsymbol{Y}(\mathcal{A})^{-1}\right]\mathrm{Tr}\left[\boldsymbol{M}_u\left[\boldsymbol{Y}(\mathcal{A})+\boldsymbol{M}_u\right]^{-1}\right]}{\lambda_{\max}\left(\boldsymbol{H}\boldsymbol{H}^T\right)\lambda_{\max}\left[\boldsymbol{Y}(\mathcal{B})^{-1}\right]\mathrm{Tr}\left[\boldsymbol{M}_u\left[\boldsymbol{Y}(\mathcal{B})+\boldsymbol{M}_u\right]^{-1}\right]}.$$

Then, let $\kappa(\boldsymbol{X}) = \sigma_{\max}(\boldsymbol{X})/\sigma_{\min}(\boldsymbol{X})$ be the $\ell_2$-norm condition number with respect to inversion, where $\{\sigma_t(\boldsymbol{X})\}$ are the singular values of $\boldsymbol{X}$. Using the fact that $\lambda_t(\boldsymbol{H}^T\boldsymbol{H}) = \sigma_t^2(\boldsymbol{H})$ yields

$$\alpha(a,b) \geq \kappa(\boldsymbol{H})^{-2} \min_{\substack{\mathcal{A},\mathcal{B}\in\mathcal{P}(\mathcal{E}) \\ \mathcal{A}\subseteq\mathcal{B},\ u\in\mathcal{E} \\ \#\mathcal{A}=a,\ \#\mathcal{B}=b}} \frac{\lambda_{\min}\left[\boldsymbol{Y}(\mathcal{B})\right]}{\lambda_{\max}\left[\boldsymbol{Y}(\mathcal{A})\right]} \times \frac{\mathrm{Tr}\left[\boldsymbol{M}_u\left[\boldsymbol{Y}(\mathcal{A})+\boldsymbol{M}_u\right]^{-1}\right]}{\mathrm{Tr}\left[\boldsymbol{M}_u\left[\boldsymbol{Y}(\mathcal{B})+\boldsymbol{M}_u\right]^{-1}\right]}. \tag{23}$$

To proceed, recall from Lemma (1) that $\boldsymbol{Y}^{-1}$ is a decreasing set function, so that $\mathcal{A}\subseteq\mathcal{B} \Rightarrow \left[\boldsymbol{Y}(\mathcal{A})+\boldsymbol{M}_u\right]^{-1} \succeq \left[\boldsymbol{Y}(\mathcal{B})+\boldsymbol{M}_u\right]^{-1}$. Since $\boldsymbol{M}_u \succeq 0$, the last term in (23) is lower bounded by one, giving

$$\alpha(a,b) \geq \kappa(\boldsymbol{H})^{-2} \min_{\substack{\mathcal{A},\mathcal{B}\in\mathcal{P}(\mathcal{E}) \\ \mathcal{A}\subseteq\mathcal{B} \\ \#\mathcal{A}=a,\ \#\mathcal{B}=b}} \frac{\lambda_{\min}\left[\boldsymbol{Y}(\mathcal{B})\right]}{\lambda_{\max}\left[\boldsymbol{Y}(\mathcal{A})\right]}, \tag{24}$$

which no longer depends on $u$, i.e., on which experiment is added to the design. We now remove the constraint $\mathcal{A}\subseteq\mathcal{B}$, which increases the feasible set and therefore reduces the value of the right-hand

side of (24). By also using the fact that $\lambda_{\min}\left[\boldsymbol{Y}(\mathcal{X})\right] \geq \lambda_{\min}\left[\boldsymbol{Y}(\emptyset)\right] = \lambda_{\min}\left[\boldsymbol{R}_\theta^{-1}\right]$ for every $\mathcal{X} \in \mathcal{P}(\mathcal{E})$, we can eliminate the dependence on $\mathcal{B}$ obtaining

$$\alpha(a,b) \geq \kappa(\boldsymbol{H})^{-2} \min_{\substack{\#\mathcal{A}=a \\ \#\mathcal{B}=b}} \frac{\lambda_{\min}\left[\boldsymbol{Y}(\mathcal{B})\right]}{\lambda_{\max}\left[\boldsymbol{Y}(\mathcal{A})\right]} \geq \frac{\kappa(\boldsymbol{H})^{-2} \lambda_{\min}\left[\boldsymbol{R}_\theta^{-1}\right]}{\max_{\#\mathcal{A}=a} \lambda_{\max}\left[\boldsymbol{Y}(\mathcal{A})\right]}. \tag{25}$$

Finally, the lower bound in (13) is obtained using Weyl's inequality to get $\lambda_{\max}\left[\boldsymbol{Y}(\mathcal{A})\right] \leq \lambda_{\max}\left[\boldsymbol{R}_\theta^{-1}\right] + \sum_{e\in\mathcal{A}} \lambda_{\max}\left[\boldsymbol{M}_e\right]$ and letting $\sum_{e\in\mathcal{A}} \lambda_{\max}\left[\boldsymbol{M}_e\right] \leq a\ell_{\max}$. $\qquad\square$

*Proof.* Start by defining the perturbed gain as $\Delta_\epsilon = \mathrm{Tr}\left[\boldsymbol{H}\boldsymbol{Y}(\mathcal{X})^{-1}\bar{\boldsymbol{M}}_u\left(\boldsymbol{Y}(\mathcal{X}) + \bar{\boldsymbol{M}}_u\right)^{-1}\boldsymbol{H}^T\right]$, for $\epsilon > 0$, where $\bar{\boldsymbol{M}}_u = \boldsymbol{M}_u + \epsilon\boldsymbol{I} \succ 0$. We omit the dependence on $\mathcal{X}$ and $u$ for clarity. Note that, $\Delta_\epsilon \to \Delta$ as $\epsilon \to 0$. Using the circular commutation property of the trace and the invertibility of $\boldsymbol{Y}(\mathcal{X})$ and $\bar{\boldsymbol{M}}_u$, we obtain

$$\Delta_\epsilon = \mathrm{Tr}\left[(\boldsymbol{H}^T\boldsymbol{H})\boldsymbol{Z}\right], \tag{26}$$

where $\boldsymbol{Z} = \boldsymbol{Y}(\mathcal{X})^{-1}\left(\boldsymbol{Y}(\mathcal{X})^{-1} + \bar{\boldsymbol{M}}_i^{-1}\right)^{-1}\boldsymbol{Y}(\mathcal{X})^{-1}$. Notice that (26) is a product of two PSD matrices, so that we can use the bound from [34] to obtain

$$\lambda_{\min}(\boldsymbol{H}^T\boldsymbol{H})\,\mathrm{Tr}(\boldsymbol{Z}) \leq \Delta_\epsilon \leq \lambda_{\max}(\boldsymbol{H}^T\boldsymbol{H})\,\mathrm{Tr}(\boldsymbol{Z}). \tag{27}$$

Let us proceed by bounding $\mathrm{Tr}(\boldsymbol{Z})$. To do so, notice that $\boldsymbol{Y}(\mathcal{A})^{-1} \succ 0$, its square-root $\boldsymbol{Y}(\mathcal{A})^{-1/2}$ is well-defined and unique [16]. We can therefore use the circular commutation property of the trace to get

$$\mathrm{Tr}(\boldsymbol{Z}) = \mathrm{Tr}\left\{\boldsymbol{Y}(\mathcal{X})^{-1}\left[\boldsymbol{Y}(\mathcal{X})^{-1/2}\left(\boldsymbol{Y}(\mathcal{X})^{-1} + \bar{\boldsymbol{M}}_i^{-1}\right)^{-1}\boldsymbol{Y}(\mathcal{X})^{-1/2}\right]\right\}. \tag{28}$$

Since (28) depends again the product of PSD matrices, we can reapply the spectral bound from [34] and obtain

$$\lambda_{\min}\left[\boldsymbol{Y}(\mathcal{X})^{-1}\right]\mathrm{Tr}\left[\boldsymbol{Y}(\mathcal{X})^{-1}\left(\boldsymbol{Y}(\mathcal{X})^{-1} + \bar{\boldsymbol{M}}_i^{-1}\right)^{-1}\right] \leq \mathrm{Tr}(\boldsymbol{Z}) \leq$$
$$\leq \lambda_{\max}\left[\boldsymbol{Y}(\mathcal{X})^{-1}\right]\mathrm{Tr}\left[\boldsymbol{Y}(\mathcal{X})^{-1}\left(\boldsymbol{Y}(\mathcal{X})^{-1} + \bar{\boldsymbol{M}}_i^{-1}\right)^{-1}\right]. \tag{29}$$

Reversing the manipulations used to get to (26) and combining (27) and (29) finally yields

$$\lambda_{\min}(\boldsymbol{H}^T\boldsymbol{H})\,\lambda_{\min}\left[\boldsymbol{Y}(\mathcal{X})^{-1}\right]\mathrm{Tr}\left[\bar{\boldsymbol{M}}_u\left[\boldsymbol{Y}(\mathcal{A}) + \bar{\boldsymbol{M}}_u\right]^{-1}\right] \leq \Delta_\epsilon(\mathcal{X}) \leq$$
$$\leq \lambda_{\max}(\boldsymbol{H}^T\boldsymbol{H})\,\lambda_{\max}\left[\boldsymbol{Y}(\mathcal{X})^{-1}\right]\mathrm{Tr}\left[\bar{\boldsymbol{M}}_u\left[\boldsymbol{Y}(\mathcal{X}) + \bar{\boldsymbol{M}}_u\right]^{-1}\right]. \tag{30}$$

The inequalities in (22) are obtained from (30) by continuity as $\epsilon \to 0$. $\qquad\square$

*Proof of Theorem 4.* This proof follows from a homotopy argument, i.e., we define a continuous map between the increments at $\mathcal{A}$ and $\mathcal{B}$ in (11) and bound its derivative. The inequality in (14) follows from applying bounds on the spectrum of Hermitian matrices.

Let $\Delta_u(\mathcal{X}) = \lambda_{\max}\left[\boldsymbol{K}(\mathcal{X})\right] - \lambda_{\max}\left[\boldsymbol{K}(\mathcal{X} \cup \{u\})\right]$ be the gain of adding $u$ to $\mathcal{X}$. Then, for $\mathcal{A}, \mathcal{B} \in \mathcal{P}(\mathcal{E})$, $\mathcal{A} \subseteq \mathcal{B}$, define the homotopy

$$h_{\mathcal{AB}}(t) = \lambda_{\max}\left[\boldsymbol{H}\boldsymbol{Z}(t)^{-1}\boldsymbol{H}^T\right] - \lambda_{\max}\left[\boldsymbol{H}\left(\boldsymbol{Z}(t) + \boldsymbol{M}_u\right)^{-1}\boldsymbol{H}^T\right] \tag{31}$$

with $t \in [0,1]$ and $\boldsymbol{Z}(t) = \boldsymbol{Y}(\mathcal{A}) + t\left[\boldsymbol{Y}(\mathcal{B}) - \boldsymbol{Y}(\mathcal{A})\right]$. Note that $h_{\mathcal{AB}}(0) = \Delta_u(\mathcal{A})$ and $h_{\mathcal{AB}}(1) = \Delta_u(\mathcal{B})$. Thus, if $\dot{h}(t)$ is the derivative of $h$ with respect to $t$, it is ready that $\Delta_u(\mathcal{B}) = \Delta_u(\mathcal{A}) + \int_0^1 \dot{h}_{\mathcal{AB}}(t)dt$. Using the definition of $\epsilon$ in (11) then yields

$$\epsilon(a,b) = \max_{\substack{\mathcal{A},\mathcal{B}\in\mathcal{P}(\mathcal{E}) \\ \mathcal{A}\subseteq\mathcal{B},\ u\in\mathcal{E} \\ \#\mathcal{A}=a,\ \#\mathcal{B}=b}} \int_0^1 \dot{h}_{\mathcal{AB}}(t)dt. \tag{32}$$

In the sequel, we proceed by upper bounding $\dot{h}$, thus getting the bound in (14). We omit the dependence on $\mathcal{A}$ and $\mathcal{B}$ for conciseness. First, to find the derivative of (31), recall from matrix analysis that $\frac{d}{dt}\boldsymbol{X}(t)^{-1} = -\boldsymbol{X}(t)^{-1}\dot{\boldsymbol{X}}(t)\boldsymbol{X}(t)^{-1}$ and $\frac{d}{dt}\lambda_{\max}[\boldsymbol{X}(t)] = \boldsymbol{u}(t)^T\frac{d}{dt}\dot{\boldsymbol{X}}(t)\boldsymbol{u}(t)$, with $\boldsymbol{u}(t)$ the eigenvector relative to the maximum eigenvalue of $\boldsymbol{X}(t)$ [32]. Then,

$$\begin{aligned}
\frac{d}{dt}\lambda_{\max}[\boldsymbol{H}\boldsymbol{Z}(t)^{-1}\boldsymbol{H}^T] &= \boldsymbol{u}(t)^T\left[\frac{d}{dt}\boldsymbol{H}\boldsymbol{Z}(t)^{-1}\boldsymbol{H}^T\right]\boldsymbol{u}(t) \\
&= -\tilde{\boldsymbol{u}}(t)^T\boldsymbol{Z}(t)^{-1}\dot{\boldsymbol{Z}}(t)\boldsymbol{Z}(t)^{-1}\boldsymbol{Z}^T\tilde{\boldsymbol{u}}(t)
\end{aligned}$$

where $\tilde{\boldsymbol{u}}(t) = \boldsymbol{H}^T\boldsymbol{u}(t)$ and $\boldsymbol{u}(t)$ is the eigenvector for the maximum eigenvalue of $\boldsymbol{H}\boldsymbol{X}(t)^{-1}\boldsymbol{H}^T$. Thus,

$$\begin{aligned}
\dot{h}(t) &= \tilde{\boldsymbol{w}}(t)^T\left[\boldsymbol{Z}(t)+\boldsymbol{M}_u\right]^{-1}\left[\boldsymbol{Y}(\mathcal{B})-\boldsymbol{Y}(\mathcal{A})\right]\left[\boldsymbol{Z}(t)+\boldsymbol{M}_u\right]^{-1}\tilde{\boldsymbol{w}}(t) \\
&\quad - \tilde{\boldsymbol{v}}(t)^T\boldsymbol{Z}(t)^{-1}\left[\boldsymbol{Y}(\mathcal{B})-\boldsymbol{Y}(\mathcal{A})\right]\boldsymbol{Z}(t)^{-1}\tilde{\boldsymbol{v}}(t),
\end{aligned} \quad (33)$$

where $\tilde{\boldsymbol{v}}(t) = \boldsymbol{H}^T\boldsymbol{v}(t)$, $\tilde{\boldsymbol{w}}(t) = \boldsymbol{H}^T\boldsymbol{w}(t)$, and $\boldsymbol{v}(t)$ and $\boldsymbol{w}(t)$ are the eigenvectors relative to the maximum eigenvalues of $\boldsymbol{H}\boldsymbol{Z}(t)^{-1}\boldsymbol{H}^T$ and $\boldsymbol{H}\left[\boldsymbol{Z}(t)+\boldsymbol{M}_u\right]^{-1}\boldsymbol{H}^T$ respectively. To upper bound (33), start by noticing that since $\boldsymbol{Y}(\mathcal{A}) \preceq \boldsymbol{Y}(\mathcal{B})$, the second term in (33) is negative. Then, using the Rayleigh's inequality yields

$$\dot{h}(t) \leq \lambda_{\max}\left[\left(\boldsymbol{Z}(t)+\boldsymbol{M}_u\right)^{-1}\left(\boldsymbol{Y}(\mathcal{B})-\boldsymbol{Y}(\mathcal{A})\right)\left(\boldsymbol{Z}(t)+\boldsymbol{M}_u\right)^{-1}\right]\|\tilde{\boldsymbol{w}}(t)\|_2^2. \quad (34)$$

We now find a bound for (34) that does not depend on $t$, so that we can apply (32). First, given that $\boldsymbol{w}(t)$ is a unit-norm vector, $\|\tilde{\boldsymbol{w}}(t)\|_2^2 = \boldsymbol{w}(t)^T\boldsymbol{H}\boldsymbol{H}^T\boldsymbol{w}(t) \leq \lambda_{\max}(\boldsymbol{H}\boldsymbol{H}^T) = \sigma_{\max}^2(\boldsymbol{H})$, where $\sigma_{\max}(\boldsymbol{H})$ is the maximum singular value of $\boldsymbol{H}$. Then, note that $\boldsymbol{Z}(t) \preceq \boldsymbol{Z}(0) = \boldsymbol{Y}(\mathcal{A})$. Thus, using the fact that for $\boldsymbol{A}, \boldsymbol{B} \succeq 0$ it holds that $\lambda_{\max}(\boldsymbol{A}\boldsymbol{B}\boldsymbol{A}) = \sigma_{\max}^2(\boldsymbol{A}\boldsymbol{B}^{1/2}) \leq \sigma_{\max}^2(\boldsymbol{A})\sigma_{\max}^2(\boldsymbol{B}^{1/2}) = \lambda_{\max}^2(\boldsymbol{A})\lambda_{\max}(\boldsymbol{B})$ yields

$$\dot{h}(t) \leq \sigma_{\max}(\boldsymbol{H})^2\lambda_{\min}\left[\boldsymbol{Y}(\mathcal{A})+\boldsymbol{M}_u\right]^{-2}\lambda_{\max}\left[\boldsymbol{Y}(\mathcal{B})-\boldsymbol{Y}(\mathcal{A})\right],$$

Thus, from (32),

$$\epsilon(a,b) \leq \max_{\substack{\mathcal{A},\mathcal{B}\in\mathcal{P}(\mathcal{E}) \\ \mathcal{A}\subseteq\mathcal{B},\ u\in\mathcal{E} \\ \#\mathcal{A}=a,\ \#\mathcal{B}=b}} \frac{\sigma_{\max}(\boldsymbol{H})^2\lambda_{\max}\left[\boldsymbol{Y}(\mathcal{B})-\boldsymbol{Y}(\mathcal{A})\right]}{\lambda_{\min}\left[\boldsymbol{Y}(\mathcal{A})+\boldsymbol{M}_u\right]^2}, \quad (35)$$

The inequality in (14) is obtained using $\lambda_{\max}\left[\boldsymbol{Y}(\mathcal{B})-\boldsymbol{Y}(\mathcal{A})\right] \leq \sum_{e\in\mathcal{B}\backslash\mathcal{A}}\lambda_{\max}(\boldsymbol{M}_e) \leq (b-a)\ell_{\max}$ for $\#\mathcal{A}=a$ and $\#\mathcal{B}=b$ and $\lambda_{\min}\left[\boldsymbol{Y}(\mathcal{A})+\boldsymbol{M}_u\right] \geq \lambda_{\min}\left[\boldsymbol{R}_\theta^{-1}\right]$. $\qquad\square$