[Reviews · NeurIPS 2017]

Reviewer 1



I believe that the authors have missed an important reference: [1] “Guarantees for Greedy Maximization of Non-submodular Functions with Applications”, by Andrew An Bian, Joachim M. Buhmann, Andreas Krause, and Sebastian Tschiatschek, ICML 2017. Given the results in [1] (in particular, see section 4.1 therein), this paper became significantly less novel. Currently I can see a few improvements of this paper over [1]: 1) The result of the additive approximate supermodularity (Theorem 2), and its corresponding consequence in E-optimal designs (Theorem 4), seems to be new. 2) The results in this paper apply to general covariance matrix Rθ, whereas [1] considers only isotropic covariance. 3) The notions of approximate supermodularity (or rather of approximate submodularity in [1]) are slightly different, but the proof techniques are essentially the same. Overall, this paper still contains nontrivial extensions, but it is really not at the level of NIPS. I therefore suggest reject. Some minor suggestions: 1) The related work is given at the end of the paper which confuses the reader. It’s better to put it at the end of the introduction. 2) Some corrections on the language usage is necessary, for example: - line 110: surrogate of - line 142: should be close to - line 166: it is worth - line 413 (at appendix): it suffices

Reviewer 2



This paper defines two types of approximate supermodularity, namely alpha-supermodularity and epsilon-supermodularity, both of which are extensions of the existing supermodularity definition. Then the authors show that the objectives of finding the A- and E-optimal Bayesian experimental design respectively belong to those two approximate supermodularity. Therefore, the greedy selection algorithm can obtain a good solution to the design problem that is within (1-e^{-1}) of the optimal, which is guaranteed by the standard result of supermodularity minimization. This paper is nicely written and everything is easy to understand. I really enjoy reading it. The result gives a theoretical justification of the effectiveness of the greedy algorithm in finding A- and E- optimal design, which is further confirmed by experiments. However, I also see some limitations of the proposed framework: 1. The analysis seems to apply only to Bayesian experimental design that the theta must have a prior. If not for the regularization term in the equation 4, it might be impossible to use any greedy algorithm. On the contrary, existing approach such as [10] does not have such limitation. 2. The relative error is different from the traditional definition, as the objective function f is normalized that f(D) <= 0 for all D. I suspect this is the key that makes the proof valid. Will this framework be applied to the unnormalized objective? Although there are limitations, I still think the paper is above the acceptance threshold of NIPS.

Reviewer 3



This work provides performance guarantees for the greedy solution of experimental design problems. Overall, this is a good paper and followings are detail comments. 1 In line 128-129, the condition should be for all A\subset B and for all u\in E\setminus B. 2 In equation (5), it is better to say e \in \argmax than e = \argmax since \argmax may return more than 1 element. 3 For Theorem 1, have you seen this paper? Bian, Andrew An, et al. "Guarantees for Greedy Maximization of Non-submodular Functions with Applications." arXiv preprint arXiv:1703.02100(2017). They got similar results. You may want to cite their works. 4 In figure, why are A-optimality and E-optimality positive? In lemma 1 and theorem 3, it is claimed than the objective is decreasing and normalized. Did you reverse the axis?